# Study and Analysis of Interference Signals of the LTE System of the GNSS Receiver

**DOI:** 10.3390/s21144901

**Published:** 2021-07-19

**Authors:** Lucjan Setlak, Rafał Kowalik

**Affiliations:** Department of Avionics and Control Systems, Faculty of Aviation Division, Military University of Aviation, 08-521 Deblin, Poland; r.kowalik@law.mil.pl

**Keywords:** interference signals, study and analysis, LTE system, GNSS receiver

## Abstract

Sometimes, it is impossible to conduct tests with the use of the GNSS system, or the obtained results of the measurements made differ significantly from the predicted accuracy. The most common cause of the problems (external factors, faulty results) are interference disturbances from other radio telecommunication systems. The subject of this paper is to conduct research, the essence of which is an in-depth analysis in the field of elimination of LTE interference signals of the GNSS receiver, that is based on the developed effective methods on counteracting the phenomenon of interference signals coming from this system and transmitted on the same frequency. Interference signals are signals transmitted in the GNSS operating band, and unwanted signals may cause incorrect processing of the information provided to the end-user about his position, speed, and current time. This article presents methods of identifying and detecting interference signals, with particular emphasis on methods based on spatial processing of signals transmitted by the LTE system. A comparative analysis of the methods of detecting an unwanted signal was made in terms of their effectiveness and complexity of their implementation. Moreover, the concept of a new comprehensive anti-interference solution was proposed. It includes, among others, information on the various stages of GNSS signal processing in the proposed system, in relation to the algorithms used in traditional GNSS receivers. The final part of the article presents the obtained research results and the resulting significant observations and practical conclusions.

## 1. Introduction 

GNSS system interference can be divided into two types: intentional and unintentional interference. Multipath interference refers to interference signals created by the reflected GNSS system signals from objects around the receiver antenna. It is one of the common sources of errors in this system. Multipath noise can alter the phase characteristics of the receiver tracking loop, leading to tracking and measurement errors. 

Research has shown that the resulting pseudo-range errors can range from a few to several hundred meters [1,2,3,4,5,6,7,8], which are sufficient to deteriorate the operational reliability of the system and positioning accuracy. However, since the effect of multipath interference shows significant differences for different observation times and station surroundings, there is currently no general and accurate mathematical model for correcting multipath errors. Therefore, suppression of multipath interference is always a hot topic of research in the field of GNSS receiver design [9,10,11]. 

Current multipath interference suppression technologies are developing primarily through the use of two approaches: receiver antenna design and signal processing. Antenna improvement technologies, from the antenna design point of view, include activities such as drawing a diagram of the multipath environment around the antenna, using special types of antennas, and selecting the appropriate locations to position the antenna [12,13,14,15]. Multipath interference suppression technologies based on signal processing fall into two main categories: time domain signal processing and spatial domain signal processing. 

Commonly used processing methods in the time domain include enhanced correlator techniques represented by the narrow correlator technique [16,17] and parameter estimation methods represented by the MEDLL technique (Multipath Estimate Delay Lock Loop) [18,19]. The process of spatial signal processing is mainly based on the fact that the LOS (line of sight) signal and multipath noise reach the receiver from different spatial directions; therefore, multipath interference suppression can be achieved using adaptive signal processing with multiple antennas [20,21,22,23,24,25,26,27,28]. 

Focusing on the problem of multipath interference suppression for the GNSS system, this article presents a model of the signal received in the event of interference from the LTE (Long-Term Evolution) system, and the impact of interference on GNSS receivers was analyzed [29]. 

An effective way of detecting irregularities in the received GPS signal can be the comparison of the determined position of the receiver with the position determined based on data from another location system, e.g., ground-based eLoran [30] or inertial systems [31,32]. The disadvantage of this solution is the higher cost (two receivers or an integrated receiver). Furthermore, the use of other navigation systems comes with limitations. The signals of terrestrial radionavigation systems may only be received within a certain area. On the other hand, in inertial systems, the problem is the accumulation of position estimation error [33]. 

The analysis of the available literature shows that currently conducted studies of this type are generally carried out under fairly favorable and established conditions, which are often manifested by the unlimited visibility of the celestial sphere. The main analysis parameter is the BER (Bit Error Rate) parameter evaluating the ratio of the correct signal to the interference signal [34,35]. 

Signals with a spread spectrum have greater resistance to fading and interference than narrowband signals with the same utility data rate. While the spectrum of the usable signal is focused in the GNSS receiver, i.e., when the signal is converted from broadband to narrowband form, the spectra of narrowband interference and selective fading are dispersed so that the signal degradation caused by them is significantly reduced. 

As a result, the influence of the first three of the above-mentioned factors is usually not so significant as to make it completely impossible to determine the position of the receiver. The issue of deliberate interference should be treated separately, where an undesirable signal, even after its spectrum is dispersed, may significantly interfere with the useful signal [36]. 

## 2. Criteria for Evaluating Solutions of Noise Elimination in a GNSS Receiver 

Both the detection methods and the methods of reducing the impact of noise coming from the GNSS receiver and their impact on the operation of the receiver can be assessed in terms of the complexity of their practical implementation and effectiveness. However, this method of assessment is very general and most often insufficient from the point of view of the recipient/user of the offered system. Then, it is necessary to make a more measurable evaluation in the form of numerical parameter values. 

Due to the very specific and innovative nature of this type of solutions, no standard set of assessment of such parameters has been established so far, so the authors of this article have attempted to define them. These parameters are universal in nature, which means that they are not only used to assess the effectiveness of a specific system analyzed in this paper but can also be used to compare other anti-noise solutions. 

The authors of this article propose the following approach to the issue of assessing the effectiveness of the noise suppression system. It should be noted that the detection of noise can be assessed by analogy to the parameters used in radiolocation, where the detection process involves the signal with the noise included and not the impulse reflected from the object as in the case of radar. The basic terms of the radiolocation theory are the probability of detection $P_{o}$ and the probability of misidentification of a PRN (Pseudo Random Noise) signal, i.e., a pseudo random signal produced by a satellite. 

The first informs about how many cases there are in which the noise will be correctly detected, and it is desirable to make it as high as possible. In turn, the second one describes how often the system will signal noise detection in the GNSS receiver, and the values of both these probabilities depend, among others, on the determined detection threshold. 

When choosing the threshold, it is necessary to make a compromise between $P_{d}$ and $P_{a}$. Depending on the desired properties, it is possible to take the lower limit of the acceptable $P_{o}$ or limit $P_{a}$ from the top. In addition to the threshold, the probability of detection is influenced by factors depending on the selected detection method. In the case of the method of comparing phase delays adopted here, the increase in the estimation error of these delays, is caused, among others, by a decrease in the SNR (signal-to-noise ratio), which is the signal processed in the GNSS receiver. 

When assessing the effectiveness of noise elimination methods, it is first of all necessary to determine to what extent this elimination affects the number of true and incomplete navigation signals and their $\frac{C}{N_{0}}$ parameter values. Noise elimination methods based on the processing of received GNSS signals should have two features. Firstly, they strongly suppress signals from external devices (LTE receivers) or separate them from useful signals. 

Secondly, it is important to enable the proper functioning of the GNSS receiver and ensure the availability of the location service based on the processed signal. In the case of the zero control method, the suppression of an unwanted signal depends on the direction of signal arrival and the estimation error of this direction, which is a function of, among others, $\frac{C}{N_{0}}$, the ratio of this signal. 

In turn, the measure of the availability of the location service is the probability of the visibility of a certain number of satellites. They can be statistically determined as the percentage of time that an anti-noise receiver is able to receive a given number of true GPS (Global Positioning System) signals, assuming there are no obstructions in the propagation paths between the satellites and the antenna of the receiver. 

### 2.1. Architecture of the Anti-Noise System for the GNSS Receiver 

The designed receiver consists of two independent receiving paths. The first track includes blocks for receiving satellite signals from a Galileo and GPS constellation in the L1/E1 band with a carrier frequency of 1.57542 GHz. The second track ensures the reception of navigation signals from both systems in the L5/E5 band with a carrier frequency of 1.17645 GHz. The entire receiving system consists of the following parts, i.e., a set of antennas that allow the reception of very weak signals from the space segment of the Galileo and GPS systems and an integrated circuit containing a heterodyne receiver, consisting of a HF (high frequency) radio path, frequency conversion system, analog IF (intermediate frequency) path, and analog-to-digital converters. 

The main task of the receiving system is to convert the signal from the antenna into a digital signal. In addition, this system consists of a digital acquisition and data processing path and final calculations responsible for decoding satellite data and performing final floating-point calculations in order to determine the geographical position, as well as a PCB (Printed Circuit Board) on which the antenna system is mounted with an integrated analog radio circuit and a digital block [37]. 

In addition to the integrated circuit, the navigation receiver system includes micro-strip antennas and discrete band-pass filters with SAW (Surface Acoustic Wave), which are designed to filter out interference. Table 1 collates the key parameters of the radio paths being the starting point for the design and production of prototypes of the GNSS receiver. 

During the first phase of the GNSS receiver project, all the above-mentioned components on three silicon structures, called the GNSS1 chipset, were designed, developed, and characterized. This division allowed for the exact characteristics and independent measurements of each block and the elimination of potential interference between successive stages of the receiving path. In the first silicon structure, a block of low-noise HF amplifiers was developed, in the second—a block of local generators and frequency transformers (baluns and mixers), and in the third integrated circuit—a block of an intermediate frequency path. In the second and final phase of the design of the GNSS receiver blocks, some of the previously made components of the radio track were redesigned, introducing appropriate corrections to improve their parameters. 

A significant progress in the project was the integration of the entire HF radio circuit on one silicon structure and transformations for the L1/E1 band and the entire HF path and transformations for the L5/E5 band in the second silicon structure, which is hereinafter referred to as the GNSS2 chipset [38]. 

### 2.2. Diagram of the Processing of Navigation Signals in a GNSS Receiver 

The system, eliminating part of the noise of the GNSS receiver, as proposed, consists of functional modules that extend the operation model of a standard GNSS receiver. To understand how a receiver with mechanisms to eliminate some noise in a GNSS receiver works, it is first needed to understand how a receiver that does not offer such a method works. 

In a standard GNSS receiver, the diagram of which is shown in the figure below (Figure 1), the signals received by the antenna are subject to band-pass filtration and are amplified. 

Most often, this process is already taking place in the active antenna itself, which is equipped with a low-noise amplifier and filters that separate the signal transmission band of one or more GNSS systems. Next, the signal is brought down to the intermediate frequency band or baseband by mixing it with the output of the local oscillator. Then, the resulting low-pass filtered signal can be further processed in analog form or converted to digital form. 

Determining the position of the receiver requires knowledge of the pseudo-range from individual satellites. The values of parameters describing the movement of these satellites along their orbits, transmitted in the form of a navigation message, are also required. In order to recover the navigation data contained in the signal of a given satellite, it is necessary to focus the spectrum of the signal, which is obtained by multiplying the input signal by the local replica. The replica is the product of the course of the C/A pseudo-random sequence and the harmonic wave with a frequency as close as possible to the central frequency of the received GPS signal. In order to produce a replica properly synchronized with the signal coming from the satellite, it is necessary to determine the parameters of this signal, which is carried out at subsequent stages of processing [39]. 

A transmission technique called DSSS (Direct Sequence Spread Spectrum) is used in GNSS systems. The reception of this type of signals is carried out in two stages. The so-called acquisition phase followed by a tracking phase can be distinguished. The first is to determine which signals are currently reaching the receiver out of all signals transmitted in a given system. In the case of the GPS system, signals are distinguished based on the forms of their C/A distracting sequences, i.e., DS-CDMA (Direct Sequence Code Division Multiple Access) code multiple access technique. 

On the other hand, the GLONASS system uses FDMA (Frequency Division Multiple Access). In addition to the identification of the signals, the Doppler deviations of the signal carrier frequency and the relative time shifts of the C/A spreading sequences are determined in the acquisition block. These are the parameters that allow replicas of signals to be produced. 

Due to the variability of these parameters over time, it is required to constantly update them in the receiver. It is the tracking block that performs the simultaneous update of the Doppler frequency and the phase of the C/A sequence. At this point, the signals are multiplied by their replicas, and carrier phase jumps 0±π due to a change in the sign of the message bit are detected. Thus, it can be concluded that focusing of the signal spectrum takes place in the tracking phase. Pseudo-distance differences from individual satellites $w^{T}$ may be determined based on the detection of bit sign change moments that define the start of a navigation message frame. The orbital parameters of the satellites and the pseudo-range information are passed to the block, calculating the position and speed of the receiver and the current system time. 

The determined navigation parameters constitute information for the presentation block, which can display them, e.g., $w^{T}$ in the form of a text or a graphic object applied to the map background. The method of implementing the anti-noise system in the form proposed by the authors requires the use of an antenna array instead of a single antenna (Figure 2). The signal from the output of each receiving antenna is frequency filtered, amplified, and frequency converted, as is done in a standard GPS receiver. 

The signals processed in this way are fed to the noise detection and spatial filtering blocks. The detection procedure determines if noise is active, and if so, which signals are the most “noisy” and what the phase delays are measured between the elements of the antenna array. The noise presence information can be used to control the position of the A and B switches. Depending on their configuration, the GPS receiver works in standard mode or in anti-noise mode [40,41]. 

The determined phase delays, as well as the signal powers from individual antennas, are transferred to the spatial filtration block, which determines the form of the weight vector, according to the Formula (1). Spatial filtration is accomplished by multiplying the vectors of the signal samples received by different antennas with the weight vector. The filtration process is described by the following Formula (1): (1)$$\begin{array}{cc} s_{fp}\left[t_{n}\right]=s\left[t_{n}\right]\cdot w^{T}\left[t_{n}\right]= & \\ =[s_{1}\left[t_{n}\right]s_{2}\left[t_{n}\right] & \ldots s_{M}\left[t_{n}\right]]\cdot \left[\begin{array}{cccc} w_{1}\left[t_{n}\right] & w_{2}\left[t_{n}\right] & \ldots & w_{M}\left[t_{n}\right] \end{array}\right]{,}^{T} \end{array}$$
where $s_{fp}\left[t_{n}\right]$ is the signal sample value after spatial filtration; $s_{M}\left[t_{n}\right]$ is the signal sample value from the output of the m-th antenna element; $w_{M}\left[t_{n}\right]$ is the current value of the m-th weighting factor, and $a^{T}$ means transposition. 

If the anti-noise system is an integral part of the GNSS receiver, as shown in the next figure (Figure 3), the signal from the spatial filtering block output is directly passed to the signal acquisition and tracking blocks. It is also possible to implement the anti-noise system as an independent system. In such a case, the output of the spatial filter is converted back to analog form and transferred to the L1 frequency band. In this form, it can be given at the HF entrance. (external antenna input) of any GNSS receiver. 

The structure of the spoofing detection block, which is the first module of the anti-spoofing system, is shown in the figure above. First, it performs the acquisition of GPS signals, which is carried out in the same way as in a standard receiver. The acquisition is based on the signal from the output of only one element of the antenna array. 

Later in this work, it was assumed that this is the first antenna element whose phase center is the reference point for measuring the phase delay of signals reaching other elements. The signal tracking block in the anti-noise system is also very similar to its counterpart in a standard receiver.

The complete tracking procedure is performed only for the signal received by the first antenna element. However, the reconstructed replica of this signal multiplies not only itself but also the outputs of the remaining array elements. This makes it possible to determine the carrier wave phases of all signals received by all antenna elements and, consequently, their phase delays and the differences of these delays between the signals modulated by different C/A sequences. The phase delay differences are the input to the actual noise interference detection algorithm. 

The decision to detect interference noise is made based on a comparison of the phase delay differences with a threshold value, depending on the number of received signals and their quality. Noise originating from LTE system signals is judged when the measured phase delay differences from at least four satellites are less than a predetermined threshold. The largest number of satellites for which the detection criterion is satisfied is considered to be the number of false signals transmitted by the base station of the LTE system. The phase delays from signals considered to be erroneous are averaged and passed to the spatial filtration block. 

## 3. Mathematical Model of Noise Removal from the LTE System 

In order to investigate effective methods of counteracting interference (unwanted LTE signals), at the very beginning, it is necessary to establish a mathematical model of the received signal, in which there are also other disturbances, including multipath, and to analyze their influence on the GNSS signal. In turn, to simplify the mathematical considerations, the GPS signal was used as an example, and the related conclusions can be generalized to other GNSS systems. Thus, the broadcast signal of the GNSS system can be mathematically represented as (2) [42]: (2)$$s\left(t\right)=D\left(t\right)c\left(t\right)\cos\left(\omega_{c}t\right),\;$$
where D(t) is the navigation message of the GNSS signal; c(t) represents C/A code; $\omega_{c}$ is the carrier frequency. 

Assuming that the GPS receiver receives the unwanted signal from the LTE system combined with multipath interference from P, the reflection paths, then the received signal can be represented as (3): (3)$$\overset{\bar{¯}}{x}\left(t\right)=\overset{P}{\underset{p=0}{\sum }}{\tilde{\alpha }}_{p}D\left(t-\tau_{p}\right)c\left(t-\tau_{p}\right)\cos\left(\omega_{c}\left(t-\tau_{p}\right)+{\tilde{\varphi }}_{p}\right)+\overset{N-1}{\underset{k=0}{\sum }}x_{k,l}\phi_{k}\left(t_{LTE}-lT\right),$$
where ${\tilde{\alpha }}_{p},$ $\tau_{p},$ ${\tilde{\varphi }}_{p}$ represent the amplitude of the beacon signal, code delay, and start phase for the p-th multipath of the GNSS signal; *p* ¼ 0 represents the LOS signal arriving directly on the unwanted signal of the LTE system, assuming that p=0 represents the SLTE signal (unwanted signal of the LTE system) arriving on the direct path, being a useful signal (information) to noise floor ratio. 

In the above Equation (3), N denotes the number of subcarrier signals present in the signal reaching the GNSS receiver, while $x_{k,l}$ denotes the number of constellation variables of the OFDM (Orthogonal Frequency Division Multiplexing) modulation signal, where k is a real number and l is an imaginary number, $\phi_{k}$ denotes an OFDM modulating signal, $t_{LTE}$ is an index of OFDM modulation prefix, T is the length of the symbol in the OFDM modulating signal, and l is the length of the block in which OFDM signals are transmitted. 

On the other hand, since the navigation message period is much longer than the C/A code period, the length of the data block is required in signal analysis by the GNSS receiver through a subsequent code delay, in which the estimation is relatively short (only a few code periods are needed C/A). During the duration of the data, the navigation message $D(t-\tau_{p})=\pm 1$, and assuming no data bit change, can further combine the value $D(t-\tau_{p})$ at (3) within an amplitude ${\tilde{\alpha }}_{p}$, and the new variable can be denoted as $\alpha_{p}^{\prime }$. Then, (3) is transformed to the form (4): (4)$$\overset{\bar{¯}}{x}\left(t\right)=\overset{P}{\underset{p=0}{\sum }}\alpha_{p}^{\prime }c\left(t-\tau_{p}\right)\cos\left(\omega_{c}\left(t-\tau_{p}\right)+{\tilde{\varphi }}_{p}\right)\cos\left(\omega_{LTE}\left(t_{LTE}-\tau_{LTE}\right)+{\tilde{\varphi }}_{k}\right),\;$$
where $\omega_{LTE}$, $t_{LTE}$, and $\tau_{LTE}$ mean the frequency, time, and delay LTE system signals, respectively, and ${\tilde{\varphi }}_{k}$ represents the phase for the p-th multipath signal of the LTE system. 

To determine the Doppler frequencies for signals arriving along different paths, consider the Doppler frequency of the SLTE signal. Hence, the following Equation (5) can be obtained: (5)$$\omega_{c}\tau_{0}=\omega_{c}\left(\frac{R_{0}\left(t\right)}{c}+\frac{L_{0}\left(t\right)}{c}\right),\;$$
where in Formula (5), the following symbols have been adopted: the speed of light is denoted by c, and the expression $R_{0}\left(t\right)=r_{0}+v_{0}t$ defines the distance between the GNSS system receiver and the signal n received from the satellite in the time sequence t, in which the distance of the measurement made in the initial stage of observation received by the receiver is defined by the signal from the satellite $r_{0}$, and $v_{0}$ defines the relative speed occurring during the transmission of the signal from the satellite. 

Furthermore, $L_{0}\left(t\right)=l_{0}+v_{LTE}t$ is the distance between the GNSS receiver and the LTE (4G) system base station; t are $l_{0}$ represent the distance between the receiver and the base station at the initial observation time, and $v_{LTE}$ is the relative rate of signal transmission between the receiver and the base station. 

The above Formula (5) can be further expressed as (6): (6)$$\omega_{c}\tau_{0}=\omega_{c}\left(\frac{r_{0}+v_{0}t}{c}+\frac{l_{0}+v_{LTE}t}{c}\right)=2\pi \left(\frac{r_{0}}{\lambda }+\frac{v_{0}t}{\lambda }+\frac{l_{0}}{\lambda }+\frac{v_{LTE}t}{\lambda }\right).$$

In the case where the relative velocity $v_{0}$ and $l_{0}$ are constant, the Doppler frequency is (7): (7)$$\omega_{d0}=2\pi \frac{v_{0}+l_{0}}{\lambda }.$$

The propagation distance of the LTE system signal can be expressed as (8): (8)$$R_{p}\left(t\right)=r_{0}+v_{0}t+l_{0}+v_{LTE}t+\Delta R_{p}\left(t\right)$$
where $\Delta R_{p}\left(t\right)$ is the propagation distance difference between the multipath interference and the LTE system signal. 

In a situation where the distance between the reflection point and the receiver is large, the code delay between the multipath interference and the SLTE signal is greater than 1.5 times the chip; then, the effect of the multipath interference on the receiver can be neglected. 

Since the satellite is far from the receiver on Earth, assuming the multi-way reflection points are not too far from the receiver, it can be assumed that $\Delta R_{p}\left(t\right)$ does not change in a short time. 

Then, $\omega_{c},\;\tau_{p}$ can be further expressed as (9): (9)$$\omega_{c}\tau_{p}=\omega_{c}\frac{r_{0}+v_{0}t+l_{0}+v_{LTE}t+\Delta R_{p}}{c}=2\pi \left(\frac{r_{0}}{\lambda }+\frac{v_{0}t}{\lambda }+\frac{l_{0}}{\lambda }+\frac{v_{LTE}t}{\lambda }+\frac{\Delta R_{p}}{\lambda }\right).$$

It can be seen that the p-th Doppler multipath interference $\omega_{dp}=\frac{2\pi v_{LTE}}{\lambda }$, i.e., the SLTE signal and the multipath interference approximately have the same Doppler frequency, and they can be determined using a unified representation $\omega_{d0}$. Hence, (4) can be further simplified as (10): (10)$$\overset{\bar{¯}}{x}\left(t\right)=\overset{P}{\underset{p=0}{\sum }}\alpha_{p}^{\prime }c\left(t-\tau_{p}\right)\cos\left[\left(\omega_{c}+\omega_{d0}\right)t+\varphi_{p}^{\prime }+\varphi_{k}^{\prime }\right],\;$$
where $\phi_{0}^{\prime }=\varphi_{0}^{\prime }+\frac{2\pi r_{0}}{\lambda }$ is the phase of the SLTE signal, $\varphi_{p}^{\prime }=\varphi_{0}^{\prime }+\Delta \varphi_{p}$, (p=1,2,…,P) is the phase of the p-th multipath signal, $\Delta \varphi_{p}=\frac{2\pi R_{0}}{\lambda }$ is the additional phase difference between the p-th multipath signal and the GNSS signal, and $\varphi_{k}^{\prime }$ represents the estimated carrier phase of the SLTE signal. 

### Influence of Other Undesirable Signals

Assuming that the carrier wave recovery is completely accurate, the signal after PLL (Phase-Locked Loop) demodulation can be represented as (11): (11)$$\bar{x}\left(t\right)=\overset{P}{\underset{p=0}{\sum }}\alpha_{p}^{\prime }c\left(t-\tau_{p}\right)\cos\left(\varphi_{p}^{\prime }-{\hat{\varphi }}_{0}^{\prime }-\varphi_{k}^{\prime }\right),\;$$
where $\;{\hat{\varphi }}_{0}^{\prime }$ is the phase shift of the carrier signal of the GNSS system, and $\varphi_{k}^{\prime }$ represents the estimated carrier phase of the SLTE signal. 

If the time interval between the early code correlator and the late code correlator is d, then the locally generated early and late codes can be represented as (12), respectively [43]: (12)$$\begin{array}{c} s_{E}\left(t\right)=c\left(t-{\hat{\tau }}_{0}-d/2\right)\\ s_{L}\left(t\right)=c\left(t-{\hat{\tau }}_{0}+d/2\right) \end{array}$$
where ${\hat{\tau }}_{0}$ represents the estimated code delay of the SLTE signal. 

After demodulation, correlations between signal and locally generated early/late codes $R_{E}\left(\varepsilon \right)$ and $R_{L}\left(\varepsilon \right)$ can be represented as (13), respectively: (13)$$\begin{array}{c} R_{E}\left(\varepsilon \right)=\overset{P}{\underset{p=0}{\sum }}\alpha_{p}^{\prime }R\left(\varepsilon +\Delta \tau_{p}-d/2\right)\cos(\varphi_{p}^{\prime }-\varphi_{k}^{\prime })\\ R_{L}\left(\varepsilon \right)=\overset{P}{\underset{p=0}{\sum }}\alpha_{p}^{\prime }R\left(\varepsilon +\Delta \tau_{p}+d/2\right)\cos(\varphi_{p}^{\prime }-\varphi_{k}^{\prime }) \end{array}\;$$
where R(·) represents the autocorrelation function of the C/A code; $\varepsilon =\tau_{0}-{\hat{\tau }}_{0}$ is an error in estimating the code delay of the SLTE signal; $\Delta \tau_{p}=\tau_{p}-\tau_{0}$ is the relative code delay between the p-th multipath interference and the SLTE signal. 

The code-tracking loop discrimination function of the GNSS receiver may be represented as (14).
(14)$$\begin{array}{cc} f\left(\varepsilon \right) & =R_{E}\left(\varepsilon \right)-R_{L}\left(\varepsilon \right)\\ & =\overset{P}{\underset{p=0}{\sum }}\alpha_{p}^{\prime }[R\left(\varepsilon +\Delta \tau_{p}-\frac{d}{2}\right)-R\left(\varepsilon +\Delta \tau_{p}+\frac{d}{2}\right)]\cos\left(\varphi_{p}^{\prime }-\varphi_{k}^{\prime }\right), \end{array}\;$$

Then, at the stage of implementing the PRN code tracking loop, in the absence of a disturbing signal (coming from the LTE system) and when there is no multipath signal influx, i.e., when the parameter ε=0 occurs, tracking errors appear in zero values code in the discrimination function. It should be noted that if the value of the discriminant function is zero, then the correlation function derived from the PRN code association values (early and late) are identical, except that the correlation function obtained from the quick code at any intermediate point obtains its maximum value. 

Assuming that the locally produced C/A code in the code tracking process is synchronized with the received code from the satellite by the GNSS system receiver, the stage of local C/A code phase tuning is performed by analyzing the zero values of the functions obtained in the discriminator. This situation provides the same correlation value for early and late codes, which means that the tracing error zeros correspond to the discrimination function zeros. 

When considering the situation related to the occurrence of undesirable signals, one should take into account their influence in the process of determining the correlation function. At this stage, mainly multipath signals arriving at the input of the GNSS receiver should be taken into account. Thus, when a condition occurs where the value of the function obtained in the discriminant is zero, and the correlation functions obtained from the early and late codes have the same waveforms, then the correlation function obtained from the quick codes is located on intermediate values and has the same waveform as the correlation function for the codes early and late. 

Bearing in mind that both the quick code correlation value and the fact that the function is in an intermediate position, hence the maximum of the correlation function is shifted from its maximum value. A condition must be met in which the delay of the signal code is zero, so it is not aligned with the position of the corresponding bit in the code. 

Assume the case that the distorted course of the correlation function contributes to the appearance of zero values in the discriminant correlation function, which results in reducing errors in the trace process. Then, it can be concluded that the situation of occurrence of zeros in the function is caused by the occurrence of a disturbing signal from the LTE system; therefore, f(ε)=0 takes place, assuming that ε is a tracking error resulting from the ongoing process of receiving an interference signal from the LTE system. 

In the case where only the SLTE signal is present, the corresponding error for the zero point of intersection of the phase discriminator function is ε=0. The DLL (Delay Locked Loop) blocks the signal by tracing the zero point of the phase discriminator function. However, when multipath noise is received, due to the effects from interference of the LTE system, the zero point of the phase discriminator function deviates from the SLTE signal code delay. When the receiver receives only one multipath interference, whereby the discriminant function in f(ε)=0, the corresponding DLL error can be derived as (15): (15)$$\varepsilon =\{\begin{array}{cc} \frac{\Delta \tau_{1}\cos\left(\varphi_{1}^{\prime }-\varphi_{k}^{\prime }\right)}{\alpha_{0}^{\prime }/\alpha_{1}^{\prime }+\cos\left(\varphi_{1}^{\prime }-\varphi_{k}^{\prime }\right)} & 0<\Delta \tau_{1}\leq \tau_{L}\\ \frac{d}{\alpha_{0}^{\prime }/\alpha_{1}^{\prime }}\cos\left(\varphi_{1}^{\prime }-\varphi_{k}^{\prime }\right) & \tau_{L}<\Delta \tau_{1}\leq \tau_{H}\\ \frac{\cos\left(\varphi_{1}^{\prime }-\varphi_{k}^{\prime }\right)}{\alpha_{0}^{\prime }/\alpha_{1}^{\prime }-\cos\left(\varphi_{1}^{\prime }-\varphi_{k}^{\prime }\right)}\left(d/2+T_{c}-\Delta \tau_{1}\right) & \tau_{H}<\Delta \tau_{1}\leq T_{c}+d/2\\ 0 & \Delta \tau_{1}>T_{c}+d/2 \end{array}$$
where $T_{c}$ is the length of the integrated circuit; then, $\tau_{L}$ and $\tau_{H}$ can be represented as (16): (16)$$\{\begin{array}{c} \tau_{L}=\frac{\alpha_{0}^{\prime }/\alpha_{1}^{\prime }+\cos\left(\varphi_{1}^{\prime }-\varphi_{k}^{\prime }\right)}{\alpha_{0}^{\prime }/\alpha_{1}^{\prime }}d\\ \tau_{H}=\frac{d\cos\left(\varphi_{1}^{\prime }-\varphi_{k}^{\prime }\right)}{2\alpha_{0}^{\prime }/\alpha_{1}^{\prime }}+T_{c}-d/2 \end{array}\;.$$

## 4. Simulation Studies 

Protection of the GNSS receiver against interference signals requires the implementation of coupled noise detection and elimination algorithms. The simplest and the least effective detection methods are based on the analysis of time dependencies and parameters related to the power of the received signals. 

For example, variations in the received signal strength during movement of the receiver are observed, or the relative delay between the signals transmitted at different frequencies is measured. In other methods, both the sample distribution of the maximum of the correlation function at the receiver and the shape of the function are monitored. Unusual indications of these parameters may indicate the presence of higher noise in the GNSS receiver. 

A more efficient way of detecting interference signals is to compare the position determined from the signals of the GPS system with the position calculated by the receiver of another navigation system. However, it should be taken into account that the signals of another radio navigation system may be disturbed, and if it is a terrestrial system, its range is limited territorially. 

Another effective way to check whether the received GNSS signals are more “noisy” could be to introduce their cryptographic protection while maintaining backward compatibility with the receivers produced so far. The signals broadcast by the satellites reach the receiver from different directions. In turn, the base station of the LTE system transmits all signals through one antenna, which means that they have the same direction of arrival. Thus, the detection of at least four signals with similar directions of arrival may indicate the presence of interference signals. 

The designed integrated system partially eliminating noise from the GNSS receiver was programmatically implemented in the Matlab/Simulink programming environment and was subjected to simulation tests. The simulation studies consisted of assessing the positioning accuracy using the spatial orientation system, GPS receiver, and the Galileo system. 

The generated errors of the integrated system elements are expressed in the local coordinate system (LCS). The values of the parameters of the GPS system model were adopted on the basis of observations of the actual measurement errors of the GNSS receiver. 

The interference signal of the LTE system is interfering with the GNSS receiver, using the correlation outputs (processes performed by the correlator in the GNSS receiver), so in this case, both stages will be analyzed: first, in the acquisition phase, i.e., that the GPS receiver cannot acquire satellites or acquire bad satellites, second, in the tracking and computing steps, the signal from the LTE system may force the GNSS receiver to output bad navigation data or not be able to demodulate the data because the LTE system signal may confuse the GNSS receiver into not finding the preamble. A genuine GNSS signal can be expressed as (17): (17)$$r_{s}\left(t\right)=\overset{M}{\underset{m=1}{\sum }}\sqrt{2P_{m}}d_{m}\left(t-\tau_{m}\right)c_{m}\left(t-\tau_{m}\right)\cos\left(2\pi \left(f_{L1}+f_{m}\right)t+\theta_{m}\right)+\overset{N-1}{\underset{k=0}{\sum }}x_{k,l}\phi_{k}\left(t_{LTE}-lT\right),$$
where $P_{m}$ is the amplitude of the satellite signal m; $c_{m}$ is the PRN code of the satellite m; $d_{m}$ is the navigation message of the satellite; $\tau_{m}$ is broadcast time; and $f_{m}$ is the frequency of the Doppler shift. 

The interference signal can be written as (18): (18)$$i\left(t\right)=Ac\left(t-\tau_{\mathrm{inter}}\right)\cos\left(2\pi \left(f_{L1}+f_{\mathrm{inter}}+f_{LTE}\right)t+\theta_{\mathrm{inter}}+\theta_{\mathrm{LTE}}\right),$$
where c(t) is the PRN code, A is the amplitude of the interference signal of the LTE system, and $\tau_{\mathrm{inter}}$ is the time shift of the interference signal. 

In the above Equation (18), parameters $f_{L1}$, $f_{\mathrm{inter}}$, and $f_{LTE}$ mean the frequency of the GPS signal transmitted in the L1 band, the interference signal frequency (GSM UMTS), and the frequency of the LTE signal, respectively. In turn, $\theta_{\mathrm{inter}}$ and $\theta_{\mathrm{LTE}}$ denote the phase of the interference signal and phase of the LTE signal. 

In fact, the authentic signal and the signal of the LTE system will be mixed upstream of the receiving antenna of the GNSS receiver. However, in the presented simulation, the real signal of the GPS system is a digital IF (Intermediate Frequency) signal, so the LTE system signals will be self-processed in the RF (Radio Frequency) front-end and then added to the real GNSS signal. In the receiver of the radio part R-front end, the complex IF signal can be expressed as follows (19): (19)$$\begin{array}{cc} r_{IF}\left(t\right)= & \overset{M}{\underset{m=1}{\sum }}\sqrt{2P_{m}}d_{m}\left(t-\tau_{m}\right)c_{m}\left(t-\tau_{m}\right)\cos\left(2\pi \left(f_{IF}+f_{\mathrm{dopp}}\right)t+\theta_{m}\right)\\ & +Ac_{\mathrm{inter}}\left(t-\tau_{\mathrm{inter}}-\tau_{\mathrm{LTE}}\right)\cos\left(2\pi \left(f_{IF}+f_{\mathrm{inter}}+f_{LTE}\right)t+\theta_{\mathrm{inter}}+\theta_{\mathrm{LTE}}\right)+n\left(t\right). \end{array}$$
where $\tau_{\mathrm{inter}}$ and $\tau_{\mathrm{LTE}}$ mean the time shift of the interference signal and LTE system, respectively. 

Then, the IF signal is passed on for secondary processing: acquisition, tracking, and computing. At a later stage, one satellite signal, e.g., from a satellite k, was considered. Then, when the local PRN code replica signal is aligned with the incoming IF signal, the output is described by the mathematical relationship (20): (20)$$\begin{array}{cc} r_{k}\left(t\right)= & \sqrt{2P_{k}}d_{k}\left(nT\right)\cos\theta_{k}\\ & +\int_{jT}^{(j+1)T}Ac_{k}\left(t-\tau_{\mathrm{inter}}-\tau_{\mathrm{LTE}}\right)c_{k}\left(t-\tau_{k}\right)\cos\left(\Delta ft+\theta_{\mathrm{inter}}+\theta_{\mathrm{LTE}}\right)dt, \end{array}$$
where $r_{k}\left(t\right)$ is the acquired satellite signal k;$\;c_{k}\left(t-\tau_{\mathrm{inter}}\right)$ is the PRN code of the LTE system signal; $c_{k}\left(t-\tau_{k}\right)$ is the PRN code of the local replica; Δf is the difference between the carrier frequency of the LTE system signal and the carrier frequency of the local replica. 

The quantities appearing in the first part are information from the navigation message, while the remaining part is the result of determining the correlation function determined from the locally generated replica code and the LTE system signal. Thus, the acquisition process performed in the GNSS system receiver depends on the second signal (LTE). It should be taken into account that the cross-correlation is characterized by low values between different PRN codes. For example, in a situation where one of the PRN codes has a high amplitude, this results in a new correlation function. 

Thus, in the acquisition phase, the GNSS receiver makes the decision to acquire one satellite signal by comparing the correlation peak to one predetermined threshold; hence, in the presence of an LTE system signal, it may not acquire the correct satellites due to this amplified PRN code. 

After the acquisition step, the IF signal will be processed in the tracking step based on the obtained coarse carrier frequency and the phase of the PRN code. As mentioned before, the tracking process includes carrier wave tracking and PRN code phase tracking. In this case, carrier tracking with PLL loop technology for a given satellite k will be analyzed: (21)$$I_{k}\left(j\right)=\sqrt{P_{k}}d_{k}\left(nT\right)\cos\phi_{k}+\int_{jT}^{(j+1)T}\frac{A}{2T}c_{k}\left(t-\tau_{\mathrm{inter}}-\tau_{\mathrm{LTE}}\right)c_{k}\left(t-\tau_{k}\right)\cos\left(\Delta ft+\Delta \phi \right)dt$$ or (22)$$Q_{k}\left(j\right)=\sqrt{P_{k}}d_{k}\left(nT\right)\sin\phi_{k}+\int_{jT}^{(j+1)T}\frac{A}{2T}c_{k}\left(t-\tau_{\mathrm{inter}}-\tau_{\mathrm{LTE}}\right)c_{k}\left(t-\tau_{k}\right)\sin\left(\Delta ft+\Delta \phi \right)dt.$$

As discussed previously, the discriminator used in the carrier tracking loop is based on the form of the function $\phi_{k}=\tan^{-1}(Q_{k\;}$/$I_{k})$; in the design algorithm, the phase error is minimized when $Q_{k}$ is equal to zero and $I_{k}$ is maximum. However, in Equations (21) and (22), if $Q_{k}$ is equal to zero, the computed phase error is (23):
(23)$$\phi_{k}=\tan^{-1}\left(\frac{\int_{jT}^{(j+1)T}\frac{A}{2T}c_{k}\left(t-\tau_{\mathrm{inter}}-\tau_{\mathrm{LTE}}\right)c_{k}\left(t-\tau_{k}\right)\sin\left(\Delta ft+\Delta \phi \right)dt}{\int_{jT}^{(j+1)T}\frac{A}{2T}c_{k}\left(t-\tau_{\mathrm{inter}}-\tau_{\mathrm{LTE}}\right)c_{k}\left(t-\tau_{k}\right)\cos\left(\Delta ft+\Delta \phi \right)dt}\right).$$

It should be noted that due to the interference signal of the LTE system, the phase error cannot be minimized to zero, and the carrier tracking fails, which makes it difficult to obtain an accurate carrier wave. 

The GNSS signal dataset used in the simulation presented was collected using a GNSS receiver in Deblin, Poland, where the GPS and Galileo satellites were visible. The processing of GNSS system signals was based on the following parameters, i.e., sampling frequency: 16.3676 MHz, intermediate frequency: 4.1304 MHz, and character sample format (8 bits). 

However, the software used in this case was developed in the Matlab/Simulink environment. The simulated LTE system signal is transmitted at a distance of 25 km from the receiver, with different power values, modulated in a carrier wave with frequencies L1 and E5. The transmitting power of the interference signal varies, and the propagation losses follow the Okumura model. 

The signal length of both the GPS and Galileo systems in this algorithm is at least 36 s to ensure the delivery of all message subframes in the navigation message. The LTE signal is 3 ms long in order not to interfere with the acquisition process. For example, a GPS signal includes five satellite signals: SV3, SV15, SV16, SV18, and SV19, SV21, and SV22. The interference signal of the LTE system is a series of sequences with many periods of the PRN code corresponding to satellite 22. 

Then, the IF signal will be processed in the tracking step based on the result of the acquisition. The trace output is the discriminator error and phase shift values. The phase shift value can be truncated to −1 and +1. In this case, finding the preamble is most important for decoding the navigation data, where the preamble indicates the start of the subframe. 

The bits appearing in the PRN code with the exemplary waveform [−1 1 −1 −1 1 −1 1 1 −1] cause the value of the correlation function of two perfectly synchronized bits of the navigation message preamble to be 8 or −8 if the inverted preamble is ideally located. In the tracking process performed by the GNSS system receiver, the information rate is 1000 sps, and each bit of navigation data is 20 ms long, which results in 20 times sampling. Thus, the value of the correlation varies between ±8 and ±160. 

### 4.1. Test Material and Research Methods 

The receiver, which is the test object, has 12 parallel tracking channels, which allows for the simultaneous reception of navigation information from up to 12 visible satellites. The signals received by the antenna of the receiver were supplied from the array of antennas. 

According to the manufacturer’s data, this receiver determines the position with an error not exceeding 15 m. It also has the ability to work in DGPS (Differential Global Positioning System) mode, which increases the precision of its indications to 3–5 m, and can also use WAAS/EGNOS (Wide Area Augmentation System/European Geostationary Navigation Overlay Service), thanks to which its error does not exceed 3 m. 

This receiver provides information about the current position and time with a frequency of 1 Hz. The aim of the conducted tests was to check the repeatability of the position indicated by the receiver in static and dynamic conditions in the case of reception of the signal provided by LTE system base stations. During the conducted tests, both stationary and dynamic, the same measurement set was used, which consisted of the following elements: The tested Garmin 16-HVS GPS receiver with wiring;A portable PC.

These elements were connected by means of adapted cabling, using the popular RJ 45 and RS 232 connectors. For communication between the computer and the GNSS receiver, the Hyper Terminal program, which is an application of the Windows environment, was used. 

The research on the processing of navigation signals in the correlator of the GNSS receiver along with the reception of the LTE signal was carried out twice. The first attempt was made when the receiver was placed on the roof of a residential building with a height of about 10 m, i.e., in conditions of very good visibility of the celestial sphere. On the other hand, the second attempt was carried out under less favorable conditions. The tested receiver was mounted on the roof of the car cab, parked in a partially wooded place, and surrounded by a brick building, which significantly reduced the visibility of the celestial sphere. 

In both cases, the conducted stationary tests consisted of the continuous measurement of the position by the receiver for about 120 min. The obtained test results were recorded in the computer memory in the form of text files. 

### 4.2. Reception of Interference Signals 

In recent years, much attention has been paid to the design of antenna systems with radiators located on non-planar surfaces. Cylindrical antennas that provide communication within the full angular range using only one antenna are interesting from the perspective of application in modern cellular systems. Conformal antennas can also be successfully used in communication systems with airplanes and small spacecraft due to the fact that they can be easily mounted, for example, in the wings of aircraft or on the outer surface of the fuselage. Therefore, an important issue is the development of integrated multi-beam antenna systems, the radiating elements of which are distributed on arbitrarily selected surfaces. 

A conformal array of antennas is an arrangement n of radiation sources distributed along a certain curve. An example of such an antenna system, in which point radiating sources are evenly distributed along the hatch, is shown in the drawing above (Figure 4). The geometry of such an antenna array can be described by the radius of the arc R and the angle ξ between the rays R passing through two adjacent radiating elements, which can be represented in the form (24).
(24)$$\begin{array}{c} F\left(\Theta \right)=a_{1}e^{j\phi 1}+a_{2}e^{j\left(2R\sin\left(\xi \right)\cos\left(\frac{\pi }{2}+\left(n-2\right)\xi -\Theta \right)+\phi 2\right)}+a_{3}e^{j\left(2R\sin\left(2\xi \right)\cos\left(\frac{\pi }{2}+\left(n-3\right)\xi -\Theta \right)+\phi 3\right)}+\\ +\cdots +a_{n-1}e^{j\left(2R\sin\left(\left(n-2\right)\xi \right)\cos\left(\frac{\pi }{2}+\xi -\Theta \right)+\phi n-1\right)}+a_{n}e^{j\left(2R\sin\left(\left(n-1\right)\xi \right)\cos\left(\frac{\pi }{2}-\Theta \right)+\phi n\right)} \end{array}$$
where R—radius of curvature; ξ—angle between rays R passing through two adjacent radiating sources. 

An antenna array configuration consisting of four elements was adopted in the research. Although the phase delays can be determined even with the use of two antennas, however, in order to reduce the phase ambiguity, it was decided to extend the system with two additional elements. There are known examples of implementations of four-element antenna arrays for counteracting jamming in systems. The greater the number of elements, the greater the efficiency of interference signal detection and spatial filtering. 

On the other hand, more signals require more computing power to process them. Moreover, with the limited physical dimensions of the array, placing the elements close to each other increases the coupling between them, which can affect the quality of the received signals. Apart from the number of antenna elements, their arrangement is also of great importance. Often, homogeneous arrays are used in which the distances between adjacent elements are the same. 

It should be added that both linear and planar systems as well as circular systems are popular. In the case of estimating the direction of signal arrival in two planes, it may be advantageous to use arrays with a three-dimensional configuration. The arrangement of antenna elements, adopted in this article, is presented in Figure 5. 

The elements are located at the vertices of a square with a side equal to 0.45 of the carrier wavelength of 1575.42 MHz, which corresponds to a distance of approximately 86 mm. Distances greater than half the wavelength would result in ambiguity in the measurement of phase delays, meaning that the same phase delays could occur for signals with different arrival directions. 

In addition, a margin of 0.05 wavelength was adopted here to limit the possibility of phase ambiguity due to an estimation error caused, among others, by the presence of noise in the channel. Placing all antenna elements in one plane results in the same phase delay values of signals arriving from symmetrical directions with respect to this plane. 

The presented characteristics (Figure 6) show that in the case of radiating an electromagnetic wave by such an antenna, the plane of the solid phase would not form a sphere whose center would uniquely define the antenna phase center. In this situation, each segment of the solid-phase plane has a specific center of curvature that is at different points for different azimuth and elevation angles. Therefore, the position of the antenna phase center is a function of the angles Θ and Ψ. Professional antennas have a specially designed structure to minimize this variability. 

The second factor interfering with measurements in applications requiring the highest precision is the variation of the antenna phase centre position depending on the incident wavelength. This is important when multi-frequency receivers are used, but they are not the subject of this research. 

In practical applications, this is not a problem as the elements are attached to a conductive surface that reflects the signals coming from the half-space on the side of the plane where there are no elements. The plane of the array should be parallel to the surface of the Earth, with elements placed on top, to enable the reception of GNSS signals from all directions for which the satellite-receiver interface is directly visible. 

In receivers, especially the most advanced ones, numerous techniques of detecting and eliminating interfering signals are used (e.g., taking into account only the signal that was received the earliest); however, the key factor improving signal quality is appropriate characteristics of GNSS antenna limiting the phenomenon of a multipath signal, which were obtained thanks to its specific design and the use of appropriate materials. 

In the literature, this phenomenon is referred to as the “mitigation of multipath effect”, which is in particular used in the CDMA system (Code Division Multiple Access). However, such antennas are very expensive and have relatively large dimensions, so they are basically used only for applications requiring ultra-precise measurements (e.g., GNSS reference stations). 

### 4.3. Noise in the GNSS Receiver—Identification 

In the literature [36,44], various approaches are proposed to solve the problem of detecting interference signals from radio systems. The selection of the appropriate method is dictated by the computational capabilities of the GNSS receiver. The least computationally complex are methods based on the analysis of parameters related to the power or time dependencies of the received signals. However, the efficiency of such solutions is relatively low, as such parameters can most often be defined so that their values do not differ from the values observed in the case of receiving signals from satellites. 

Another suggestion is to compare the indications of the designated position with the position determined using another navigation system; however, in this case, territorial limitations (ground-based radio navigation systems) or problematic calibration (inertial systems) should be taken into account. 

Moreover, it was postulated to introduce cryptographic safeguards to navigation messages contained in civil GPS signals. This would involve designating a digitally signed hash of the navigation message to be transmitted to verify that it is genuine. The hash would be sent in message fields that are not currently in use, thus maintaining backward compatibility with all receivers produced so far. The inconvenience of this solution is the necessity to introduce changes on the broadcasting side, which requires actions on the part of the authorities supervising the operation of the satellite system. 

One of the most effective methods of detecting radio interference signals is the analysis of the direction of the incoming signals. When signals are received from GNSS satellites in direct visibility conditions, their arrival directions to the receiving antenna are different. It is different in the case of interference where all the signals produced are transmitted using the same transmitting antenna, hence their directions of arrival being the same. 

To determine the direction of the signal arrival, it is important to use a system of several antennas (antenna array) and measure the relative phase delays of the signals reaching the individual antennas. The receiver needs to receive at least four GNSS signals for the position to be determined by the receiver. Thus, it can be concluded that spoofing is present when the phase delay values associated with at least four GPS signals are close to each other. 

#### The Dependence of C/N on $E_{b}/N_{0}$

The key issue in modeling the GNSS receiver in the situation of receiving LTE system interference signals is the determining the element dependence of the error rate BER (Bit Error Rate) at the output of the GNSS receiver as a function of the signal-to-noise ratio ($C/N_{0}$) and the signal-to-interference ratio (C/I) at the input of this decoder in the channel with additive white Gaussian noise AWGN (Additive White Gaussian Noise) as the basic one in the determination of protection factors for different LTE transmission modes (different carrier modulations and different convolutional code efficiencies). Knowledge of these functions enables computer simulations of the element error rate depending on the parameters of the useful GNSS signal, noise parameters of the transmission path, and the level of interfering signals. 

The detection threshold, i.e., the value of the signal phase delay difference below which a positive interference detection decision is made, depends on the number of false signals received and their quality, as expressed by the parameter $C/N_{0}$. The values of this parameter measured in real conditions range from 35 to 60 dBHz. The most sensitive receivers can detect signals $C/N_{0}$ as low as 30 dBHz, but after demodulation, their waveform is close to noise. 

It should be noted that the obtainable probability of detecting interference is the greater the more signals of disturbing signals are received and the greater their value $C/N_{0}$. 

In the remainder of this work, the element error rate BER will be determined as a function of the ratio $E_{b}/N_{0}$ (energy per bit to noise density). To describe the relationship between $E_{b}/N_{0}$ and the signal-to-noise ratio at the demodulator input $C/N_{0}$, the above relationship, which is true for modulated signals based on the method of bandwidth sharing with division into the frequency OFDM (Orthogonal Frequency Division Multiplexing) method with different guard interval D, symbol duration $T_{u}$ and code efficiency $R_{c}$ [44] values can be used. 

The considered dependence can be written by the following Equation (25): (25)$$C/N=\log_{2}M\frac{E_{b}}{N_{0}}R_{c}\frac{1}{1+\Delta /T_{u}}$$
where M is the number of modulator states (4 for QPSK, 16 for 16 QAM, and 64 for 64 QAM), whereby QPSK (Quadrature Phase Shift Keying) is a modulation with 2-bit coding of the transmitted signal with four orthogonal phase shifts, and QAM (Quadrature Amplitude Modulation) is amplitude-phase modulation, which is designed to transmit digital data over a radio channel. 

It should be noted that on the basis of the above dependence (25), it is possible to unequivocally determine C/N with the help of a known value $E_{b}/N_{0}$ and vice versa. This will make it possible to move from the receiver signal parameters (useful signal power and noise power) to the value $E_{b}/N_{0}$ and thus determine the BER value. 

It should be noted that it is possible to determine the interference power that will prevent the reception of any GNSS satellite signals. According to the official satellite-receiver interface specification [45], the minimum power of the GNSS signal with string C/A (generally available) at the receiver antenna output should be −160 dBW (10^−16^ W). With the bandwidth $B_{C/A}$ of the signal with string C/A, being approximately 2 MHz, the signal power to thermal noise power ratio is −19 dB. Therefore, the signal is received significantly below the noise floor. 

A measure of the quality of the received GPS signal is the ratio $C/N_{0}$ of the carrier wave power to the spectral noise power density after focusing the spectrum, which is expressed in dBHz units and can be written as follows (26): (26)$$SNR\left[\mathrm{dB}\right]=\frac{C}{N}\left[\mathrm{dB}\mathrm{Hz}\right]-10\;\log_{10}\left(B_{\frac{C}{A}}\left[\mathrm{Hz}\right]\right)=\frac{C}{N}\left[\mathrm{dB}\mathrm{Hz}\right]-63\;\left[\mathrm{dB}\mathrm{Hz}\right]$$
where the signal-to-noise ratio (SNR) in the above formula defines the signal-to-noise ratio measured by the GPS receiver. 

The threshold value $C/N_{0}$, below which a GNSS receiver is not able to receive the signal correctly depends on its sensitivity and is usually not less than 30 dBHz. This corresponds to a signal-to-noise ratio of −33 dB. Therefore, if the power of the useful signal was e.g., 10 dB greater than the minimum and was −150 dBW, in order to effectively disturb it, it would be enough to transmit a narrowband signal whose power at the receiver input would be −150 dBW + 33 dB = −117 dBW, i.e., about 2 × 10^−12^ W. 

The power of the interfering transmitter obviously depends on the length and nature of the propagation path between this transmitter and the interfered receiver. Nevertheless, a relatively low power transmitter is usually sufficient. It should also be noted that the susceptibility to interference depends on whether the receiver is tuned to the useful signal at the moment of activation of the unwanted signal (i.e., is it in the so-called tracking phase) or is just searching for it (acquisition phase) [43]. 

The research described in [34,35] indicate that a transmitter transmitting an interfering signal with a power of 244 mW, in a bandwidth of 2 MHz, is sufficient to prevent the reception of GPS signals within a radius of more than 6 km when the receiver is tuned to signals from satellites and within a radius of more than 8 km when the receiver is not yet synchronized with them. 

The effectiveness of the interfering signal of GNSS signals does not only depend on the power of the received interfering signal but also on its frequency characteristics. The spectrum of Gold’s string pulses is stripe-like, where the striations are spaced from each other and from the carrier frequency by a multiple of 1 kHz, which is the reciprocal of the period of a 1 ms pseudorandom sequence. 

The publication [42] shows that high-interference efficiency is obtained by transmitting signals whose power spectral density is high at these frequencies (e.g., mono- and poly-harmonic signals). The only difficulty in implementing the interference in this case is that the carrier frequency of the GNSS signal is shifted by the Doppler frequency, one component of which depends on the—generally unknown—velocity of the satellite relative to the receiver. 

The research described in [34,35] has also shown that strong interference outside the GNSS systems band may also have a negative impact on the quality of reception of navigation signals. The algorithms that are used to calculate the ratio of the carrier wave power to the spectral power density of the noise make it possible to make a measurable assessment of the quality of the received signals. Correct estimation of signal quality is essential to obtain system or interference signal performance characteristics. It should be noted that calculations of the $C/N_{0}$ ratio using different methods give similar but non-identical results. 

In this article, it is assumed that the estimated $C/N_{0}$ values should deviate as little as possible from the real ones in the range of 35 to 60 dBHz, because such values are observed in real conditions. Simulations were performed to calculate $C/N_{0}$ using the variance summation method, the Beaulieu method, and the moment method.

The obtained results are presented in the figure above (Figure 7). As can be seen, for values not lower than 40 dBHz, the absolute error for all three methods does not exceed 0.5 dB. The variance summation method slightly underestimates the result, and the method of moments similarly overestimates it. The smallest error is obtained in the Beaulieu method; however, for values less than 40 dBHz, it exceeds −2 dB. 

Taking into account the entire considered range from 35 to 60 dBHz, the smallest mean error is obtained when using the variance summation method. Therefore, this method was chosen for the estimation $C/N_{0}$ in this paper. 

From the results obtained, it can be estimated that the improvement is about 2 dBHz, which translates into an improvement in the obtained positioning accuracy of the GNSS receiver. 

The figure below (Figure 8) shows a part of the correlation results when searching for the location of the satellite preamble 15 in one channel. It is seen from the figures that there are some high peaks that can go up to 160 or a little less. For a 37 s long signal, six preamble patterns should be found, and each of them has 6 s distances from adjacent patterns. 

On the other hand, the figure above (Figure 9) shows exemplary implementations of positioning errors in an anti-noise system operating without GPS correction. These errors build up quickly while the system is running. 

Selected results of simulation tests of signal processing in the correlator are shown in the following figures (Figure 10, Figure 11 and Figure 12). 

The presented results confirm the effectiveness of the developed method of noise elimination in the form of interference signals. The obtained results for the GPS signal translate into an improvement in the accuracy of determining the position of a user within 1 to 2 m in terms of the coordinates x and y. It can be seen that for height, this improvement is about 0.5 m. 

Similar simulations were carried out for the Galileo system, as shown in the figures below (Figure 13, Figure 14 and Figure 15). 

The results for the Galileo system are similar to those for GPS signals. The position of a user with a Galileo receiver has improved by 1.5 to 2 m. However, the difference between the two different simulations is not due to the signal of the LTE system; the reason is that the GPS receiver selects different sets of satellites in the computation performed in the correlator. 

The operation of interference signals from the LTE system makes the position of a user less precise by “hiding” one visible satellite. More research has been done on what happens when the signal power of the LTE system becomes low and high. Studies have shown that when the power is 10 times higher than the power considered previously, the GPS receiver cannot acquire enough satellite signals (not at least four satellites but only two) to obtain position information. 

In the case of receiving the signal provided by the LTE system at the input of the GNSS system receiver, this signal may be treated as an interfering signal. The obtained results from computer simulations show that when the signal from the LTE system has a power 10 times lower than the adopted standards, it does not result in a deterioration of the accuracy in the process of determining the position by the GNSS system. 

## 5. Summary and Conclusions 

The effect of the signal from the LTE system is realized and tested to determine how this might affect the GNSS system. The main purpose of the article was to answer the question of how it will affect symbol synchronization in the PRN code or time acquisition. All phenomena were analyzed in the GNSS receiver correlator. 

The method of implementation of the filter system adopted in this paper enables strong suppression of interference signals of the LTE system arriving from one direction, which is defined by the values of phase delays of carrier waves (Figure 2). 

When a signal reaches the receiver via multiple paths, only the reception of the strongest component and possibly other signals coming from the same direction are blocked. If at least one of the other spatial components of the received signal is relatively strong, the reception of GNSS satellite signals may be hindered. 

The simultaneous elimination of signals coming from different directions is possible using such a form of the spatial filtration weights vector, in which the receiving characteristic has many zeros. If the antenna array is built of n elements, it is possible to suppress signals from up to n−1 directions (Figure 4). 

In this case, it is necessary to determine the phase delay sets independently for each component of the antenna array in such a way that such zeros are more spatially dispersed; therefore, real signals coming from other directions than those of the LTE system may also be attenuated. 

For example, Figure 7 presents a comparison of averaged values of the $C/N_{0}$ signal GNSS system and GNSS system with simultaneous reception of the LTE system signal; also plotted is a case of intentional jamming of the GNSS signal, when the signal arrives from the direction of azimuth Ψ and elevation Θ. Each of the determined values of transmission quality is an average value of 1000 results calculated for different realizations of phase delay estimation error, occurring in the GNSS receiver. It should be noted here that the scatter of this error in transmission quality depends on the $C/N_{0}$ ratio. 

When GNSS signals are received with low-quality LTE signals, they are attenuated on average by several dBHz. The use of a filtering module in this situation allows achieving an improvement in $C/N_{0}$ attenuation of unwanted signals in the range of 2 to 3 dBHz. Thus, with a high $C/N_{0}$ ratio, the total attenuation reaches 50 dBHz, which, as mentioned earlier, is the maximum attenuation value possible in this array configuration for the assumed signal arrival direction. 

The simulations performed prove that the LTE system signal may cause disturbances in the carrier wave signal and in the PRN code transmitted by the satellite. The computed position may deviate slightly from that determined in the absence of an LTE signal when the GNSS receiver selects different satellites in the computation performed by the radio part of the navigation receiver. 

In the case when the signal power of the LTE system is high enough, it can “jam” the GPS receiver in such a way that the receiver was not able to acquire enough signals from the satellites and estimate the noise value currently in the receiver, which could reduce the measured distance. 

In turn, due to the fact that the GNSS system signal uses direct carrier modulation using the DSSS (Direct Sequence Spread Spectrum) code sequence, after conversion to navigation message data bits, the error rate (number of bits incorrectly received) BER will be much lower, because 20 × 1023 corresponds to one bit of navigation data. 

In contrast, when an interference signal is introduced from an LTE system, BER depends on which part of the signal is dominant in demodulation. So, to effectively decode the GNSS signal, it should not be too large to dominate the demodulation. 

Simulations were performed for dynamic situations in realistic urban environments. Four types of GNSS measurements were considered during the simulations, including the LOS, LTE system interference signal, diffraction, and multipath, which covers most of the interference found in an urban area (Figure 7). 

From the point of view of each measurement, both the one-hour static experiment and the dynamic experiment with vehicles confirm that the measurement from the proposed simulator has a consistent and reasonable behavior in the error range compared to the real measurement. Furthermore, it should be added that the proposed algorithm effectively eliminated some of the noise from the correlator of the GNSS system receiver. 

In the next stage, the simulated measurements were applied together with advanced positioning algorithms, which verifies that the proposed simulator can adequately reflect the current difficulties in both precise positioning and in the scope of the so-called “bottleneck” of different positioning algorithms. Therefore, the proposed simulator can provide realistic GNSS system measurements for multiple intermediaries (agents) to investigate and improve state-of-the-art GNSS system positioning algorithms in an urban area. 

It should be noted that in addition to the well-known “noise” caused by the interference signal of the LTE system occurring in the navigation tracking loop, the effect resulting from the initial detection process (or correlation process) and the use of the developed method significantly contributes to reducing the systematic error resulting from the fading of the navigation signals of the PLL loop of the GNSS system receiver. 

Another result of the analysis and simulation is the phenomenon referred to as the so-called dynamic short-range effect. This effect is an error induced in the tracking loops when the receiver moves relatively close to the signal transmitter of the LTE system. 

In this case, the higher derivatives of the line of sight can reach large values, even if the receiver is moving at a constant speed. This causes range errors in the tracking loops due to their indeterminate response. Additionally, in some cases, this can even cause the receiver to lose lock, whereby when the signal comes from a satellite, this effect should not be significant. However, in the case of pseudolites, it can be significant. 

The most significant errors are mainly generated in situations where the direct line of sight of navigation signals by the GNSS receiver is blocked and the receiver is able to track an indirect signal. In theoretical considerations, this can result in a range of errors of virtually any magnitude. 

In all simulations involving complete constellations, it has been clearly demonstrated that in difficult conditions (urban canyons and mountainous areas), the combined use of GPS and Galileo can significantly increase the availability and quality of service. Therefore, interoperability between these systems (GPS and Galileo) is very important (Figure 8, Figure 9, Figure 10, Figure 11, Figure 12, Figure 13, Figure 14 and Figure 15). 

Thus, in general, it can be said that as signal structures become more sophisticated and efficient, it is more important to reduce other sources of error in the GNSS system. For example, it would be appropriate and reasonable for the Galileo to provide more accurate orbit and clock parameters than the GPS system in order to improve the overall positioning accuracy. 

## Figures and Tables

**Figure 1 sensors-21-04901-f001:**
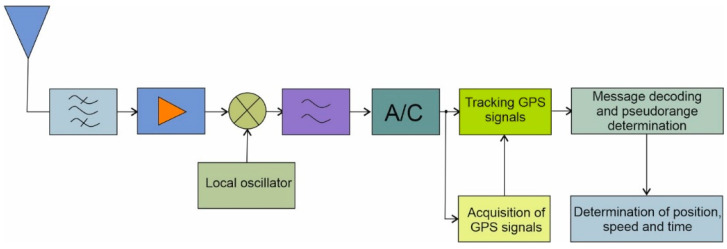
Block diagram of the GPS receiver.

**Figure 2 sensors-21-04901-f002:**
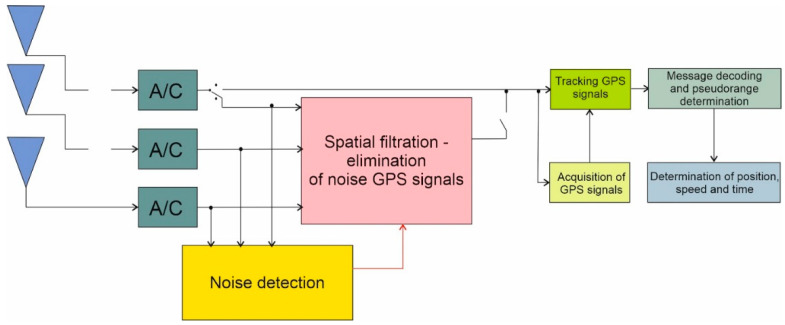
Block diagram of a GPS receiver with an anti-noise system.

**Figure 3 sensors-21-04901-f003:**
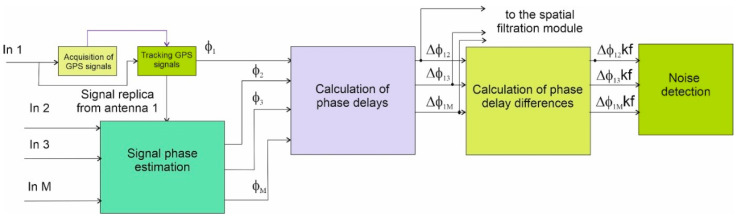
Schematic of noise detection block from the LTE signal.

**Figure 4 sensors-21-04901-f004:**
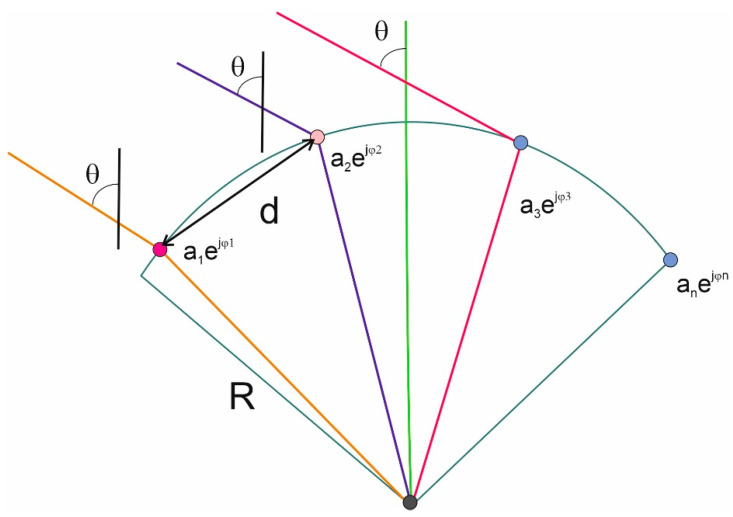
Antenna array—receiving signals in the GNSS receiver.

**Figure 5 sensors-21-04901-f005:**
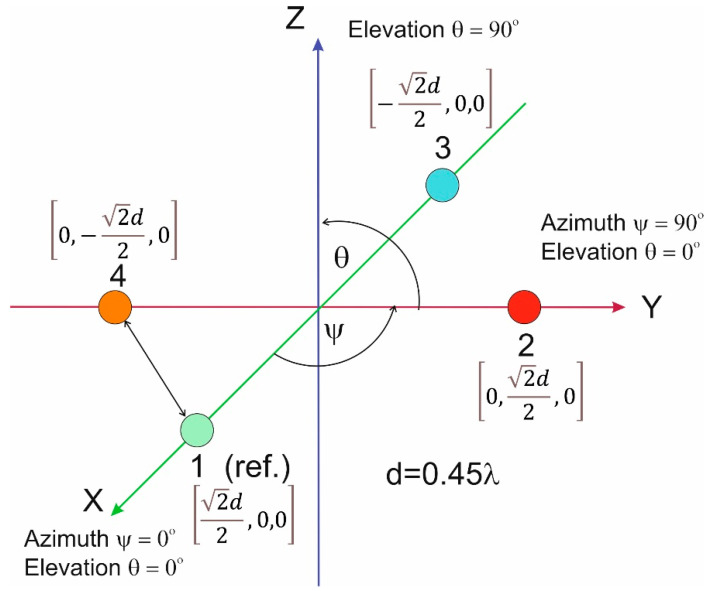
Arrangement of antennas receiving navigation signals.

**Figure 6 sensors-21-04901-f006:**
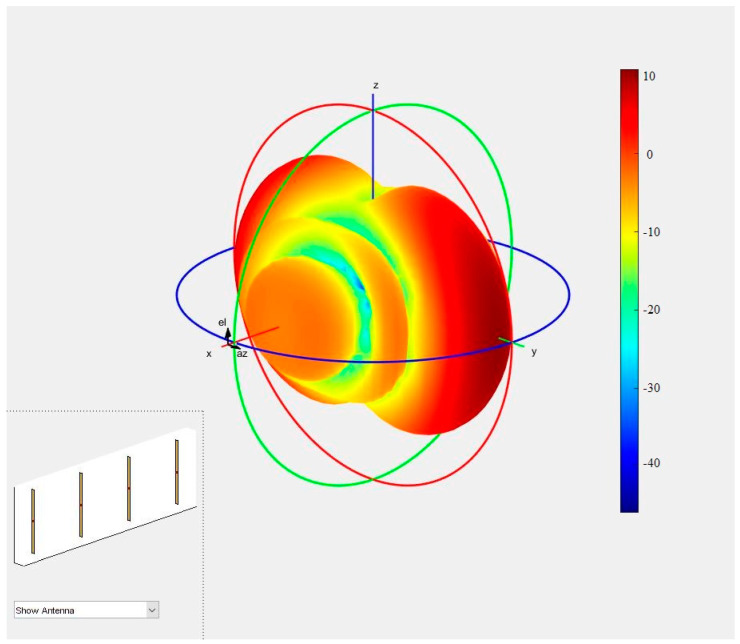
The radiation characteristics of the antenna used in the research for the 1554 MHz frequency.

**Figure 7 sensors-21-04901-f007:**
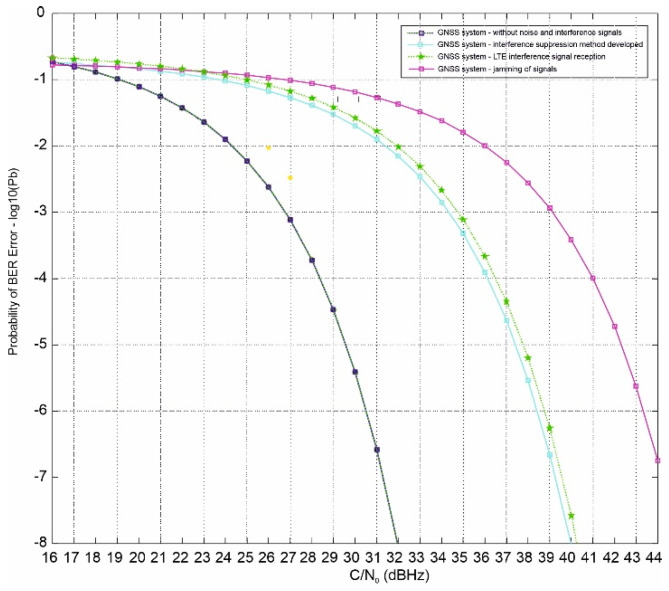
The course of the $C/N_{0}$ ratio for the GNSS system with the presence of LTE system interference signals.

**Figure 8 sensors-21-04901-f008:**
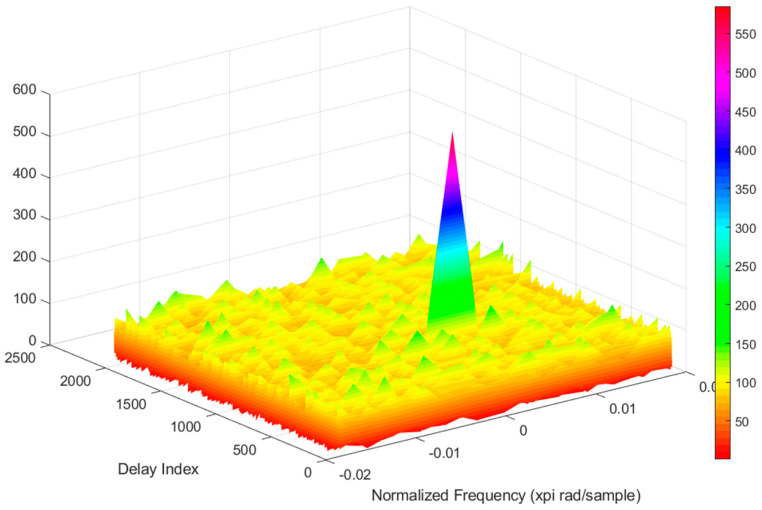
Correlation function for the GPS system signal.

**Figure 9 sensors-21-04901-f009:**
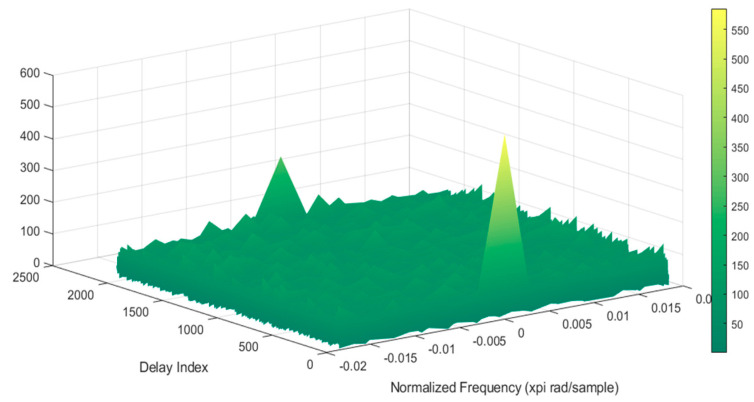
Correlation function for the Galileo system signal.

**Figure 10 sensors-21-04901-f010:**
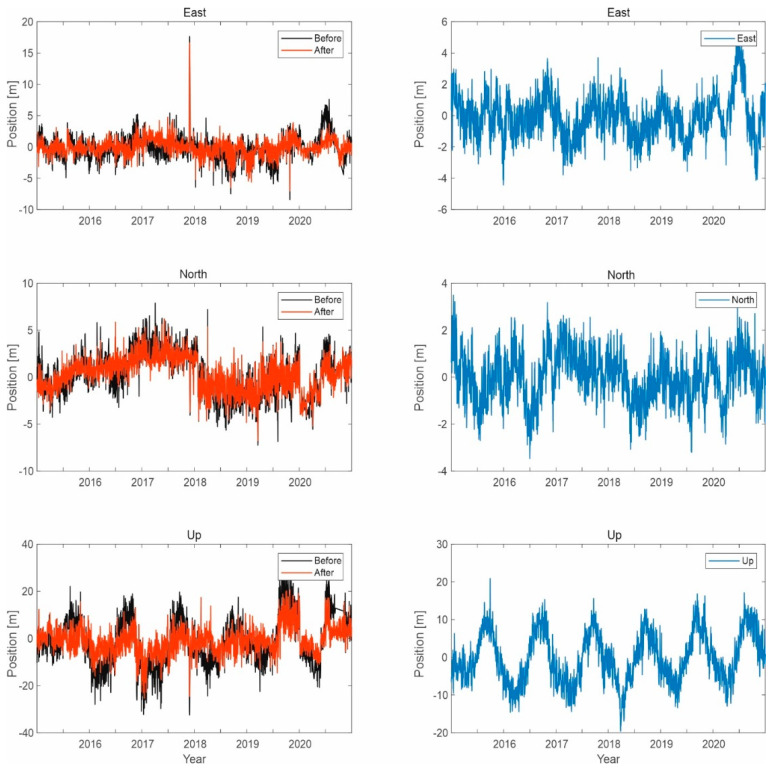
Signal processing in the GPS receiver working on the L1 frequency; on the left side, the signal waveform in color without the noise suppression system applied; on the right side, the working system operating in the LTE system signal elimination for the reception of signals from four satellites.

**Figure 11 sensors-21-04901-f011:**
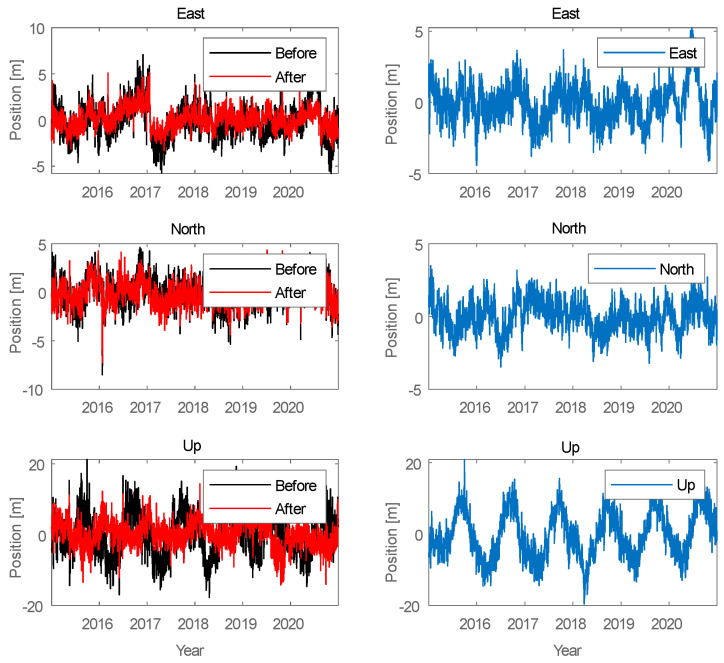
Signal processing in the GPS receiver working on the L1 frequency; on the left side, the signal waveform in color without the noise suppression system applied; on the right side, the working system operating in the LTE system signal elimination for the reception of signals from five satellites.

**Figure 12 sensors-21-04901-f012:**
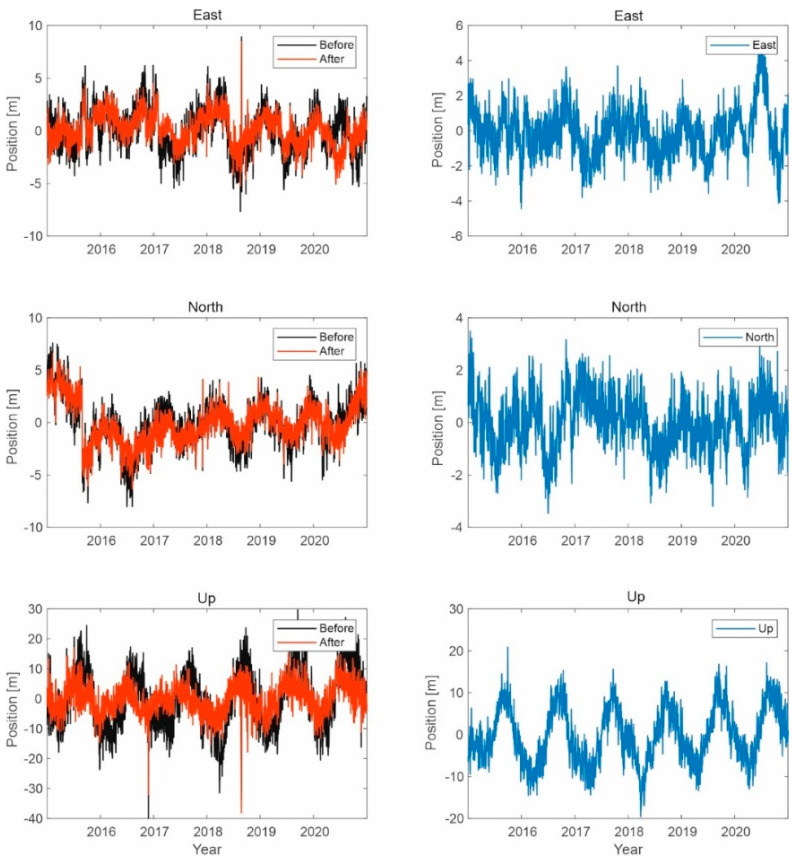
Signal processing in the GPS receiver working on the L1 frequency; on the left side, the signal waveform in color without the noise suppression system applied; on the right side, the working system operating in the LTE system signal elimination for the reception of signals from six satellites.

**Figure 13 sensors-21-04901-f013:**
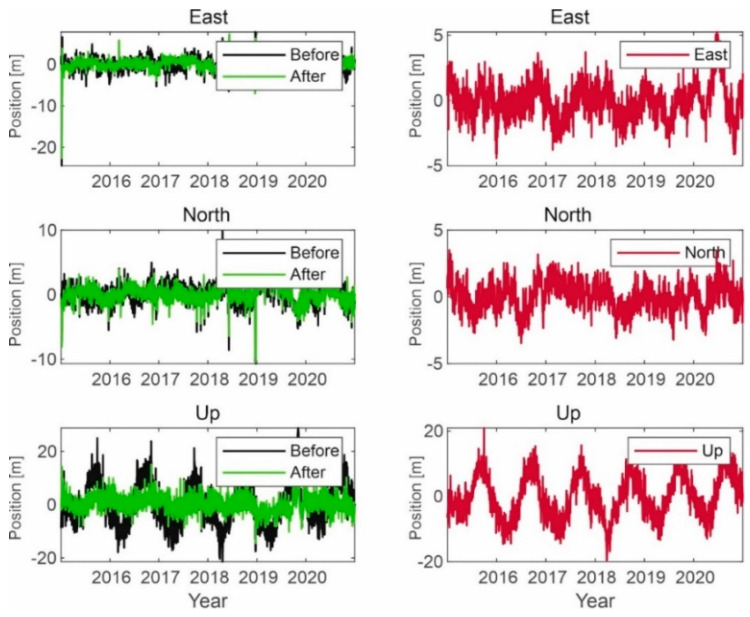
Signal processing in the Galileo receiver operating on the L1 frequency; on the left side, the signal waveform in color without the noise suppression system applied; on the right side, the working system operating in the LTE system signal elimination for the reception of signals from four satellites.

**Figure 14 sensors-21-04901-f014:**
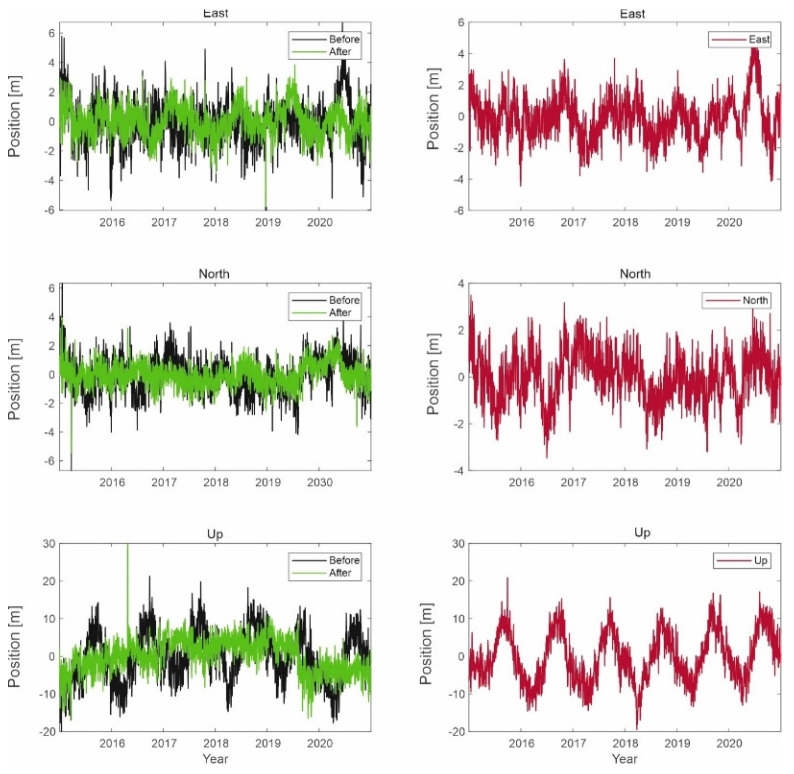
Signal processing in the Galileo receiver operating on the L1 frequency; on the left side, the signal waveform in color without the noise suppression system applied; on the right side, the working system operating in the LTE system signal elimination for the reception of signals from five satellites.

**Figure 15 sensors-21-04901-f015:**
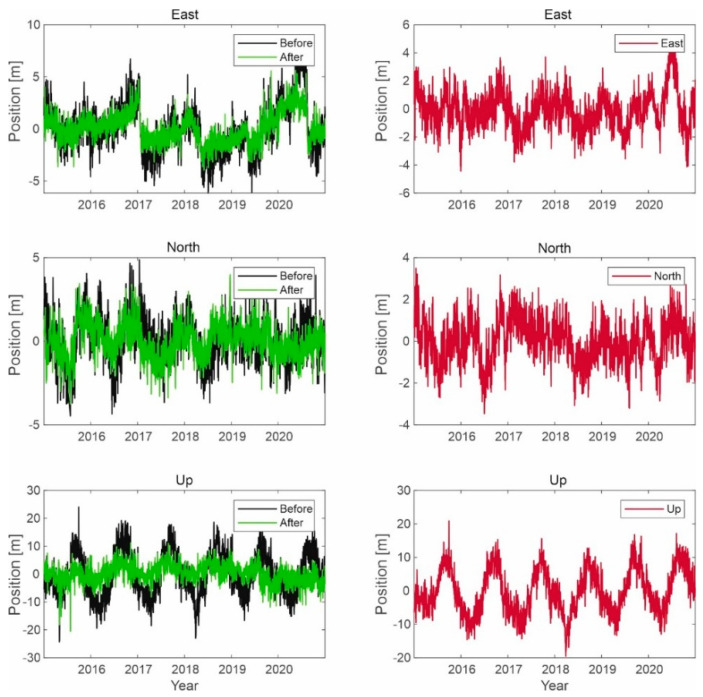
Signal processing in the Galileo receiver operating on the L1 frequency; on the left side, the signal waveform in color without the noise suppression system applied; on the right side, the working system operating in the LTE system signal elimination for the reception of signals from six satellites.

**Table 1 sensors-21-04901-t001:** Main parameters of the GNSS receiver of the Galileo constellation and GPS for the bands L1/E1 and L5/E5.

Block	Parameter	Unit	L1/E1	L5/E5
Antenna system	Directional gain	[dBi]	3	3
	Antenna band	[MHz]	5	26
High-frequency track	Amplification	[dB]	47.2	48.8
	Noise figure NF_dsb_	[dB]	1.99	1.87
	Power consumption	[mA]	4.4	4.8
Frequency synthesis	Phase noise at ΔF_C_ = 1 MHz	[dBc Hz]	−110	−90
	Power consumption	[mA]	2.8	1.8
Intermediate-frequency track	Passband of the filter	[MHz]	3–5.2	0.05–24
	Amplification	[dB]	46–81	42–77
	Mirror frequency suppression	[dB]	>40	0
	Power consumption	[mA]	3.3	7.2
Whole	Amplification	[dB]	93–128	91–126
	Noise figure NF_dsb_	[dB]	1.99	4.87
	Passband	[MHz]	3–5.2	0.05–24
	Mirror frequency suppression	[dB]	>40	0
	A/C converter resolution	[bits]	1.5	1.5
	Power consumption	[mA]	10.5	12.8
	Integrated circuit power supply	M	1.2 ± 10%	
	Operating temperature	[°C]	−40–+125	

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
