# Peer review of "Study and Analysis of Interference Signals of the LTE System of the GNSS Receiver"

_sensors, 2021, doi:10.3390/s21144901_

Round 1

Reviewer 1 Report

The manuscript proposes the use of multiple antennas at a GNSS receiver to suppress interference for LTE signals.

As far as the technical content, the manuscript provides some insight into the interference problem. The authors tried to account for the channel model and the main parameters. However, I do not see how many antennas do they consider in their numerical results. Moreover, it is not completely clear from their manuscript how do they design beamforming weights. 

The language is rather poor and must be improved by seeking the help of a native speaker or an expert.  Regarding the presentation, but the new scheme with multiple antennas is not properly introduced. Maybe a dedicated section should emphasize its role.

A clear statement that helps to understand the difference with respect to the state-of-the-art is necessary. 

Author Response

The abstract and introduction of this article have been modified according to the recommendations and comments made in the review. Special attention was paid to interference signals transmitted in the operating band of the GNSS navigation system. The article focuses on the methods of detection of interference signals, including in particular those transmitted by the LTE system, and makes their comparative characteristics in terms of efficiency and complexity. Moreover, a comprehensive approach to the problem solving of the anti-interference approach has been proposed, using a number of references from the research literature on the current state of knowledge.

Furthermore, in line with the reviewer's valuable comments, 4 subsections explaining the use of multiple antennas in a GNSS receiver to suppress interference for LTE signals have been added to Chapter 4 of this article. Other comments, e.g. improvement of the English language, have also been addressed.

Any modifications made to this article are highlighted in yellow.

Reviewer 2 Report

This paper presents an anti-noise system for GNSS receivers.

  • The title of this paper should be changed. This paper is to solve a scientific problem by analyzing and eliminating LTE interference signals of the GNSS receiver. Do you mean you want to eliminate the interference caused by the LTE systems for GNSS receivers?
  • GNSS systems are developed based on CDMA technology. The receivers have a built-in capability of anti-noise (not anti-jamming). Why do the authors want to study such a topic? Please give more motivations for this research.
  •  The block diagrams of Figs. 2 and 3 are not consistent. For example, the "Calculation of phase delays" in Fig. 3 outputs signals to the "spatial filtration module". But, in Fig. 2, no such input signals from Fig. 3. Please check it.
  • Lack of analysis of the noise characteristics caused by the LTE systems to the GNSS receivers.
  • For the anti-noise topics, simulation results have shown some improvements to your proposed method. However, it seems insignificant. Please quantify your improvements and compare them with other existing works, such as
    • O. Apilo et al., "Measured GPS performance under LTE-M in-device interference," 2018 9th ESA Workshop on Satellite NavigationTechnologies and European Workshop on GNSS Signals and Signal Processing (NAVITEC), 2018, pp. 1-7, doi: 10.1109/NAVITEC.2018.8642666.
    • Y.R. Chien, P.Y. Chen, S.H. Fang Novel anti-jamming algorithm for GNSS receivers using wavelet-packet-transform-based adaptive predictors IEICE Trans. Fundam. Electron., Commun. Comput. Sci., 100 (2) (2017), pp. 602-610

Based on my comments, the reviewer cannot suggest accepting this paper in its current form.

Author Response

The article has been revised in accordance with the reviewer's comments and recommendations in the field of research, methods used, presentation of results, their analysis and the final stage of the work (summary, relevant observations). In the aspect of valuable remarks of the reviewer, the title of the article has been modified and 4 subchapters of the research chapter have been developed, in which, among other things, the characteristics of noise caused by LTE systems for GNSS receivers have been analysed by comparing the proposed solution with other solutions in the literature of the research subject and the analysis of the obtained research results has been conducted.

Any modifications made to this article are highlighted in yellow.

Round 2

Reviewer 1 Report

The authors improved their work considerably by adding details on the multiple antenna setup. The system model now is more clear.

I noticed that in many places the ratio Eb/No is mistakenly represented as Eb=No and the same is for C/No which appears as C=No. Please check carefully.

Reviewer 2 Report

In this revision, all my previous concerns have been well-addressed. The reviewer has no further comments for this revision. Thus, the reviewer would like to suggest accepting this manuscript in its current form.
